# GAD65 deficient mice are susceptible to ethanol-induced impairment of motor coordination and facilitation of cerebellar neuronal firing

**Wataru Matsunaga**[¤a]*, **Toru Shinoe**[iD], **Moritoshi Hirono**[iD][¤b]*

RIKEN Brain Science Institute, Wako, Saitama, Japan

¤a Current address: Joint-Use Research Facilities, Hyogo Medical University, Nishinomiya, Hyogo, Japan
¤b Current address: Department of Physiology, Wakayama Medical University, Wakayama, Wakayama, Japan
* wa-matsunaga@hyo-med.ac.jp (WM); mhirono@wakayama-med.ac.jp (MH)

**Data Availability Statement:** All relevant data for the electrophysiology are within the manuscript and its Supporting information file except for the behavioral data.

## Abstract

γ-aminobutyric acid (GABA) is a major inhibitory neurotransmitter and its concentrations in the brain could be associated with EtOH-induced impairment of motor coordination. GABA is synthesized by two isoforms of glutamate decarboxylase (GAD): GAD65 and GAD67. Mice deficient in GAD65 (GAD65-KO) can grow up to adulthood, and show that GABA concentration in their adult brains was 50–75% that of wild-type C57BL/6 mice (WT). Although a previous study showed that there was no difference in recovery from the motor-incoordination effect of acute intraperitoneally administered injections of 2.0 g/kg EtOH between WT and GAD65-KO, the sensitivity of GAD65-KO to acute EtOH-induced ataxia has not been fully understood. Here, we sought to determine whether motor coordination and spontaneous firing of cerebellar Purkinje cells (PCs) in GAD65-KO are more sensitive to the effect of EtOH than in WT. Motor performance in WT and GAD65-KO was examined by rotarod and open-field tests following acute administration of EtOH at lower-doses, 0.8, 1.2 and 1.6 g/kg. In a rotarod test, there was no significant difference between WT and GAD65-KO in terms of baseline motor coordination. However, only the KO mice showed a significant decrease in rotarod performance of 1.2 g/kg EtOH. In the open-field test, GAD65-KO showed a significant increase in locomotor activity after 1.2 and 1.6 g/kg EtOH injections, but not WT. In *in vitro* studies of cerebellar slices, the firing rate of PCs was increased by 50 mM EtOH in GAD65-KO compared with WT, whereas no difference was observed in the effect of EtOH at more than 100 mM between the genotypes. Taken together, GAD65-KO are more susceptible to the effect of acute EtOH exposure on motor coordination and PC firing than WT. This different sensitivity could be attributed to the basal low GABA concentration in the brain of GAD65-KO.

**Funding:** The author(s) received no specific funding for this work.

**Competing interests:** The authors have declared that no competing interests exist.

**Abbreviations:** EtOH, ethanol; GABA, γ-aminobutyric acid; GAD, glutamate decarboxylase; WT, wild-type.

## Introduction

γ-Aminobutyric acid (GABA) is a major inhibitory neurotransmitter in the mammalian brain [1]. It also acts as a trophic factor during neural development [2]. GABA plays a crucial role in not only emotional [3, 4] but also behavioral functions, including circadian rhythm [5], locomotor activity [6], and motor coordination [7, 8]. GABA is synthesized by two isoforms of glutamate decarboxylase (GAD), GAD65 and GAD67 [9]. These isoforms are generally co-expressed in the same GABAergic neurons, where GAD65 and GAD67 are found mainly in nerve terminals and cell bodies, respectively [10, 11]. Previous studies with gene disrupted mice have shown that GAD67 is expressed at early developmental stages, whereas GAD65 develops during postnatal maturation [12–16]. GAD67 knockout mice do not survive after birth due to a cleft palate [13, 14]. On the other hand, GAD65 knockout mice (GAD65-KO) can grow up to adulthood, even though the GABA concentration in their adult brains is 50–75% that of wild-type mice (WT) [15, 17]. The KO mice show seizure susceptibility and abnormal emotional behavior [12, 15, 16, 18].

Ethanol (EtOH), a psychoactive drug, acts as a modulator of multiple neurotransmissions in the central nervous system (CNS) [19], and EtOH administration causes various behavioral disorders, such as the loss of the righting reflex [20, 21], the alterations of locomotor activity [22–24], and the impairments of motor coordination [20, 25], in a dose-dependent manner. A lot of evidence suggests that the EtOH-mediated effects are attributed to the modulation of GABAergic transmission in the CNS [26, 27], in particular, the potentiation of postsynaptic $GABA_A$ receptor functions [8, 28–31]. Acute exposure to EtOH causes impairments in motor coordination, most likely due to cerebellar dysfunction [31], which are accompanied by EtOH-induced neuronal firing facilitation in the cerebellar cortex [32–35]. The firing facilitation can be suppressed by EtOH-induced enhancement of GABAergic inhibition [32, 36–38], leading to protection from neuronal over-excitation and maintaining normal motor coordination. However, the relationship between acute EtOH-induced ataxia and basal GABA concentration in the brain has not been fully understood.

This study aimed to clarify whether motor coordination and locomotor activity in GAD65-KO are more sensitive to the effect of EtOH than in WT. We also investigated the effect of EtOH on spontaneous firing of cerebellar Purkinje cells (PCs) in GAD65-KO compared to WT.

## Materials and methods

### Animals

We used male C57BL/6 genetic background GAD65-KO (10–12-week-old) and WT littermates for the experiments. The generation of GAD65-KO was described previously [12]. Experimental mice were obtained from 9 to 12 generations made by $GAD65^{+/-} \times GAD65^{+/-}$ intercrossing, and their genotypes were determined after weaning by the polymerase chain reaction protocol (https://mus.list.brc.riken.jp/ja/wp-content/pdf/00989_PCR.pdf). The mice were kept at our animal facility under a constant temperature and a 12-h light/dark cycle (light on at 7:30 am) with free access to food and water. They were housed usually in groups of 3–5 mice per cage, with no distinction between GAD65-KO and WT. The bedding was changed once every two weeks after the experiments. All the experimental procedures were performed in strict accordance with the *Guide for the Care and Use of Laboratory Animals* described by the National Institutes of Health and adhered to the Animal Research: Reporting of *In Vivo* Experiments (ARRIVE) guidelines. These experimental procedures were approved and

overseen by the RIKEN Animal Research Committee. Every effort was made to minimize the number of animals used and their suffering.

## Drugs

We prepared three EtOH solutions at 10%, 15% and 20% (v/v) concentrations. All solutions were made fresh on an experimental day. Injections were given intraperitoneally (i.p.) at a volume adjusted according to the mice's body weight of 0.1 ml per 10 g. To prevent the influence of the solution weight difference, 10% solution was used for a 0.8 g/kg injection, 15% solution for a 1.2 g/kg injection and 20% solution for a 1.6 g/kg injection because the density of EtOH is 0.79 g/ml. The blood and brain EtOH concentrations are thought to reach the maximum level at 3 min after injection [39], so all experiments started 3 min after injection.

## Measurement of home cage activities

The circadian rhythm of home cage activity and the acute EtOH effect in the home cage were measured by the Activity Sensor System (O'Hara and Co., Tokyo, Japan). This system detects animals as an infrared image, and each shape change of the infrared image counts as animal activity. All experimental mice were caged alone in transparent polycarbonate boxes (21.5 × 13 × 15 cm) with food and water *ad libitum*. Their cage activities were measured every hour for 2 weeks, followed by 48 h of habituation in an experimental room. Once a week, activity sensors were stopped during feeding and the supply of water.

## Blood EtOH concentration

A QuantiChrom EtOH Assay Kit (BioAssay Systems, USA) was used in this study. Mice were anesthetized and blood was collected from the heart. Blood plasma samples were separated by centrifugation, and deproteinization and EtOH detection were performed following the manufacturer's recommended protocol.

## Open-field test

In this study, we used a four-channel open-field observation system (O'Hara and Co.). This system consists of four arenas, each 50 cm × 50 cm, and walls 40 cm high in a white plastic box under a 70-lux incandescent lamp. During a 30-min trial, the behavior of the mice was continuously recorded by a video camera placed over the center of the arena. Video images were stored as Tag Image File Format (.tiff) file at one flame per second, and the total moving distance was automatically calculated by video tracking software (O'Hara and Co.). At least 60 min before the beginning of each test, animals were moved into the test room. Each mouse was picked up by its tail, placed in the same corner of the arena. The tests were performed on three consecutive days. On day 1, mice were not given any injections and were placed into the arena. On day 2, mice were injected with saline and placed in the arena 3 min after injection. On day 3, mice were given one of the three doses of EtOH (0.8, 1.2 and 1.6 g/kg). All tests were performed from 1:00 pm to 3:00 pm to minimize the effect of circadian rhythm.

## Rotarod test

Balance and motor coordination were tested by using a constant-speed (20 rpm) rotarod apparatus (Model 7600 with a 3 cm in diameter rod; Ugo Basile, Comerio, Italy). One trial was a maximum 180 sec on a rotating rod and at least 3 min of rest in the home cage. The latency to fall of each trial was recorded. When the mice did not fall within 180 sec, 180 sec were allotted for their latency to fall [40]. To detect the time effect after injection strictly, each experimental

mouse was placed on the rod every 6 min. If mice fell before 180 sec, they were rested for 3 min plus the remainder of the riding time. Experimental mice were moved from the colony room at least 60 min before each test, and all tests were performed during 1:00 pm and 3:00 pm to minimize the effect of the circadian rhythm. The tests were performed on five consecutive days. On day 1, mice were injected with saline and then given five trials starting 3 min after injection. On day 2 and 3, mice were given 10 trials after saline injection. On day 4, mice were injected with 0.8 or 1.2 g/kg EtOH and given 10 trials, starting 3 min after the EtOH injection. On day 5, mice were injected with 1.2 or 1.6 g/kg EtOH and given 10 trials. All mice were given a higher dose of EtOH than the previous day, and we confirmed that there was no "hangover" effect for latency to fall at 24 h after EtOH injection in a preliminary experiment.

## Electrophysiological recordings

Cerebellar slices were prepared from WT and GAD65-KO aged 24–38 days of either sex as previously described [32]. The mice were deeply anesthetized via isoflurane inhalation and then decapitated. Sagittal slices (250-μm thick) of the cerebellar vermis were obtained using a vibrating microtome (VT1200S, Leica, Nussloch, Germany) in an ice-cold extracellular solution containing (in mM) 252 sucrose, 3.35 KCl, 21 NaHCO3, 0.6 NaH2PO4, 9.9 glucose, 0.5 CaCl2, and 10 MgCl2 and gassed with a mixture of 95% O2 and 5% CO2 (pH 7.4). The slices were maintained at room temperature for at least 1 h in a holding chamber, where they were submerged in artificial cerebrospinal fluid (ACSF) containing (in mM) 138.6 NaCl, 3.35 KCl, 21 NaHCO3, 0.6 NaH2PO4, 9.9 glucose, 2 CaCl2, and 1 MgCl2 (bubbled with 95% $O_2$ and 5% $CO_2$ to maintain the pH at 7.4). Individual slices were transferred to a recording chamber attached to the stage of a microscope (BX51WI, Olympus, Japan) and superfused with oxygenated ACSF. Recordings were performed from PCs located exclusively in lobules III–VIII to limit the variability. Spike activity in PCs was observed using loose cell-attached voltage-clamp recordings, which allowed long recordings without changing cytoplasmic content. Glass electrodes (2–3 MΩ) were filled with ACSF and gently placed in contact with PCs. Slight suction was applied, and the holding potential was set to 0 mV. Experiments were performed at room temperature (24–26 ˚C). The firing patterns of PCs are generally divided into three types: tonic, bursting, and silent [41]. We bath-applied EtOH to PCs that exhibited continuous tonic firing patterns to estimate the effect of EtOH on PC excitability upon increasing firing rate. Previous studies have reported that PCs in slices and *in vivo* fire tonically, with occasional periods of bursting and silence. In our cerebellar slice preparations, most of the PCs fired tonically, while some were silent. This may have been due to the temperatures during recordings [41] or the damage to their axon initial segments during slice cutting. All other chemicals were purchased from Sigma-Aldrich (St. Louis, MO). The membrane currents were recorded using an amplifier MultiClamp 700B (Molecular Devices, Sunnyvale, CA) and pCLAMP 10.3 software (Molecular Devices), digitized, and stored on a computer disk for offline analysis. All signals were filtered at 2–4 kHz and sampled at 5–20 kHz. Spike firing was analyzed using the Mini Analysis Program 6.0 (Synaptosoft, Decatur, GA), Clampfit 10.3 software (Molecular Devices), and KyPlot software (version 6.0; KyensLab, Tokyo, Japan). Throughout the data analyses of electrophysiological signals, *n* refers to the number of recorded neurons.

## Statistics

All data were expressed as mean ± standard error of the mean (SEM) throughout the manuscript. Behavior data except for locomotor activity mentioned below were statistically analyzed using the repeated measures two-way analysis of variance (ANOVA) or the repeated measures one-way ANOVA. Bonferroni's test was used as a post-hoc test in all cases. The results of

locomotor activity in Figs 1B, 2A and 2B were analyzed by unpaired and paired *t*-test. These analyses were performed using FreeJSTAT statistical software (version 22.0E; http://toukeijstat.web.fc2.com/). The level of significance of electrophysiological data was determined using the Wilcoxon signed-rank test or the Mann–Whitney U test for comparisons between related and independent sample groups. These analyses were performed using the KyPlot software (version 6.0; KyensLab). The spike train regularity of PCs was assessed by measuring the coefficient of variation (CV) and CV2. To acquire CV2 for the spikes in a train arise at time $t_i$ ($0 \leq i \leq$ N) inter-spike intervals (ISIs) are $ISI_i = t_i - t_{i-1}$ for $1 \leq i \leq$ N. CV2 is calculated as $CV2_i = (2|ISI_{i+1} - ISI_i|)/(ISI_{i+1} + ISI_i)$. An averaged CV2 was obtained over *i*. CV2 will be lower when the variation in a train occurs at time scales considerably longer than the typical ISI [42].

## Results

### Activity of GAD65-KO and blood EtOH concentration

We recorded the home cage activities of WT and GAD65-KO (Fig 1A). The data in Fig 1A were calculated as follows: (1) averaging the activity of each mouse every 60 minutes over the entire measurement period, (2) grouping the data from each mouse into WT and GAD65KO, and (3) averaging them. A significant difference was detected in activity counts each hour between WT and GAD65-KO by repeated measure two-way ANOVA (genotype × time block through whole day) ($F(1, 455) = 15.650$, $p < 0.001$). The curves of activity counts during the daytime showed a similar pattern for WT and GAD65-KO, and two-way ANOVA (genotype × time block in daytime) did not detect significance ($p = 0.321$). Their mean values were 149.02 ± 33.57/hour and 121.66 ± 18.02/hour, respectively. By contrast, during the nighttime, GAD65-KO showed a decrease in activity counts compared to WT, and significant difference were detected by two-way ANOVA (genotype × time block in nighttime, $F(1, 227) = 18.376$, $p < 0.001$), almost similar to a previous report [43]. Their mean values were 784.34 ± 101.08/hour and 454.43 ± 67.71/hour, respectively. Thus, we performed all behavioral experiments during the day to prevent any circadian rhythm effect.

We performed the open-field test repeatedly on each mouse to compare within the same animal and minimize animal use. A previous study with the repeated open-field test showed a significant difference in locomotor activity and other behaviors between days 1 and 2, but no significant difference after day 2 [44]. Our preliminary experiment showed no significant difference in locomotor activity between days 2 and 3. Additionally, the experience of the open-field test did not affect the EtOH-induced locomotor activity in the open-field arena [45]. Therefore, in this study, we gave a saline injection on day 2 and an EtOH injection on day 3, and we could ignore the repeat effects on our results for the open-field test. Our open-field test showed that the moving distance of GAD65-KO was significantly higher than that of WT in the case of both no-administration ($p < 0.01$) and saline injection ($p < 0.05$) (Fig 1B). Moreover, no significant difference was observed between no administration (day 1) and saline injection (day 2) in either genotype (Fig 1B). We examined EtOH concentrations in the blood for 60 min after EtOH injection prior to behavioral analyses with EtOH administration. At 10, 30 and 60 min after 1.2 g/kg EtOH i.p. injection, there was no significant difference between WT and GAD65-KO (Fig 1C), suggesting that ethanol metabolism are similar between the genotypes.

### Effects of EtOH on locomotor activity in the open-field test

In the open-field test, WT showed no significant difference between saline injection and any dose of EtOH injection (Fig 2A). On the other hand, GAD65-KO mice showed significant

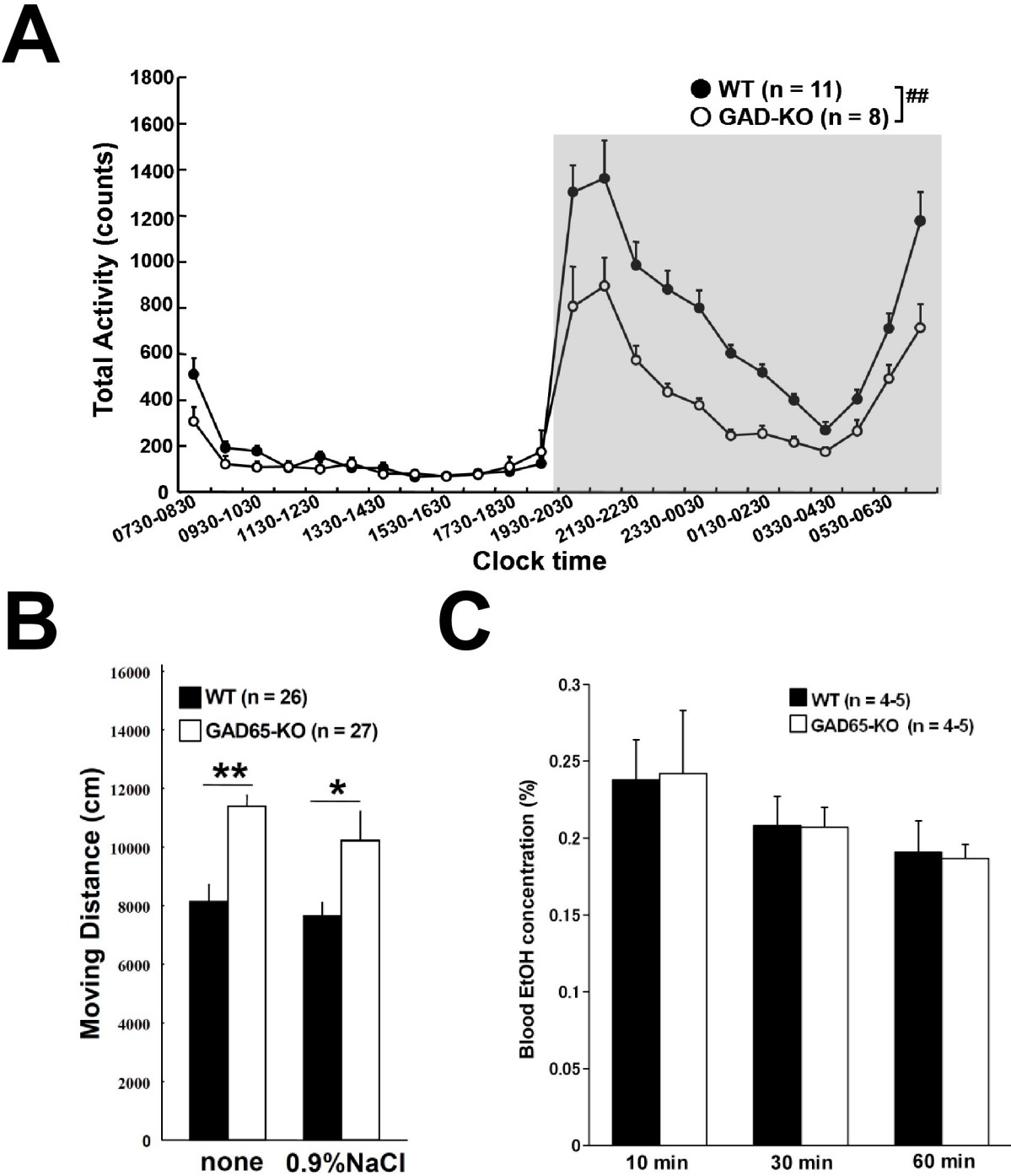

**Fig 1. Activity of WT and GAD65-KO and blood EtOH concentration.** (A) The circadian rhythm of home cage activities of WT and GAD65-KO. The data were calculated by averaging the activity of each mouse every 60 minutes over the entire measurement period and then grouping the data from each mouse into WT and GAD65-KO and averaging them. WT showed significantly higher activity than GAD65-KO during the nighttime (1930–0729, gray area). However, no significant difference was observed during the daytimes (0730–1929). (B) Moving distance in the open field. Results of the non-administration and saline-injected groups showed that the basal locomotor activity of GAD65-KO was significantly higher than that of WT mice. (C) Blood EtOH concentrations after 10, 30 and 60 min of intraperitoneal injection of 1.2 g/kg EtOH. There was no significant difference between WT and GAD65 KO (n = 4–5 in each group). $^{\#\#}p < 0.01$: significant difference between WT and GAD65-KO during the night. $^{*}p < 0.05$, $^{**}p < 0.01$: a significant difference detected by an unpaired $t$-test.

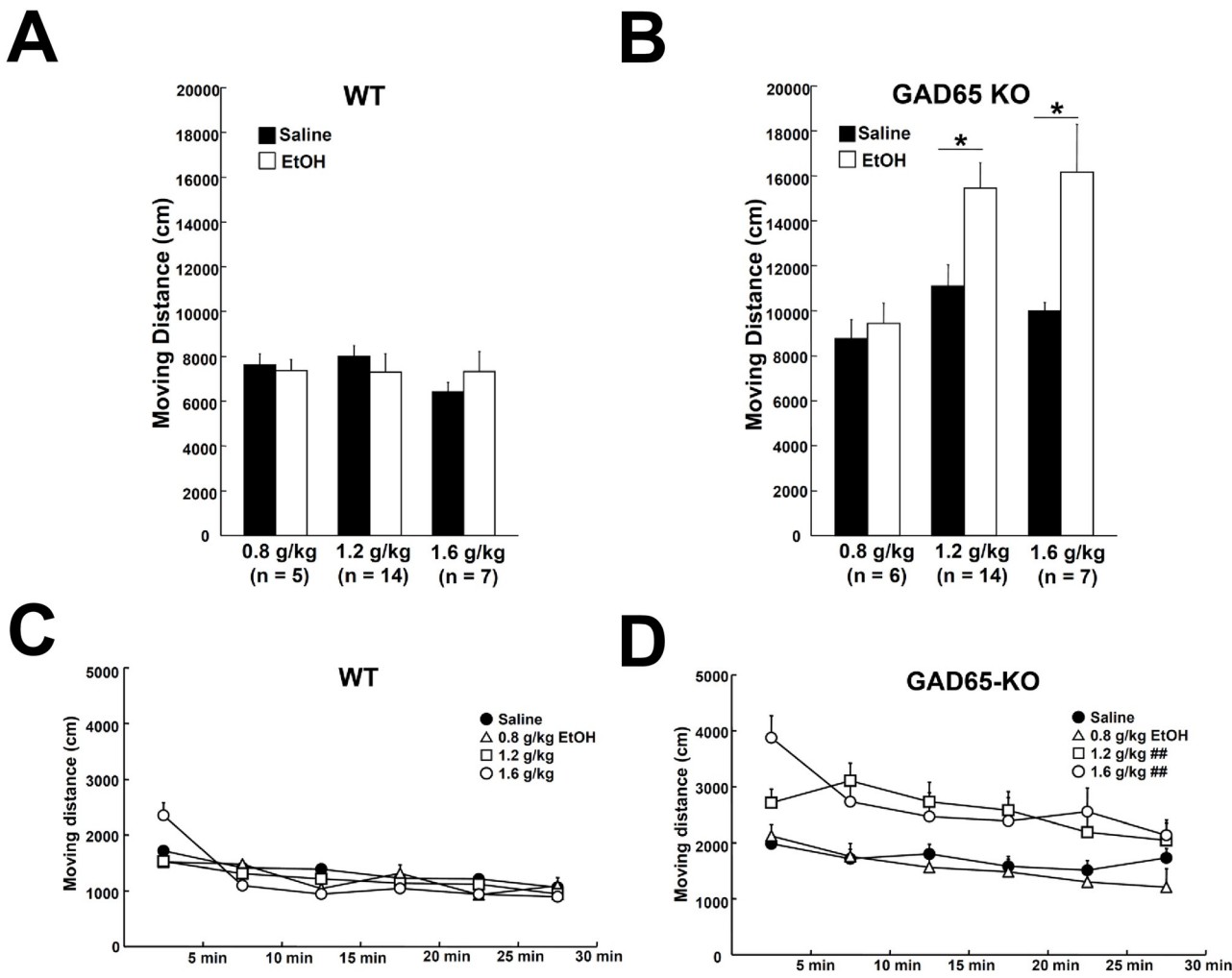

**Fig 2. Moving distances in the open-field.** (A) In WT, EtOH injection did not change the average distance moved. (B) GAD65-KO showed a significant increase in locomotor activity after 1.2 and 1.6 g/kg EtOH injections. (C and D) Averaged distance moved every 5 min by WT and GAD65-KO. At all experimental doses, EtOH injection had no effect on WT (C). GAD65-KO showed a significant increase in distance moved during all experimental periods after 1.2 and 1.6 g/kg EtOH injections, but no significant effect was observed after a 0.8 g/kg EtOH injection (D). $^*p < 0.05$: a significant difference detected by a paired $t$-test. $^{##}p < 0.01$: a significant difference in the saline-injected group detected by a two-way ANOVA.

increases in locomotor activity when injected with EtOH at 1.2 ($t = -2.963$, $p < 0.05$) and 1.6 g/kg ($t = -2.685$, $p < 0.05$) (Fig 2B).

Locomotor activity was recorded every 5 min. In WT, a repeated-measures two-way ANOVA of moving distance every 5 min (EtOH dose × time block) detected no significant difference with EtOH dose ($p = 0.2434$) (Fig 2C). On the other hand, a significant effect of EtOH dose was detected in the results of GAD65-KO (F(3, 227) = 5.136, $p < 0.01$) (Fig 2D), and the 1.2 and 1.6 g/kg EtOH-injected groups showed significantly higher locomotor activity ($p < 0.01$) with a significant increase in locomotor activity within the same time block in the saline-injected group. Moreover, a significant effect of time passage was detected in both the WT (F(5, 227) = 23.535, $p < 0.001$) and GAD65-KO (F(5, 227) = 14.812, $p < 0.001$) (Fig 2C and 2D). Regardless of the genotype or EtOH dose, all mice show a tendency to decrease locomotion with time.

## Effects of EtOH on rotarod performance

All experimental mice were habituated and trained to remain in the rotarod for 3 days prior to the EtOH injection. On day 1, the latency to fall gradually increased in both WT and GAD65-KO, suggesting a significant effect of the trial (F(4, 139) = 8.412, $p < 0.001$). The learning curves of rotarod performance did not differ between the genotypes ($p = 0.282$). On day 2, we still observed a significant effect of trials (F(9, 269) = 1.964, $p = 0.0445$) but no difference between the genotypes ($p = 0.187$). On day 3, regardless of genotype, almost every mouse successfully rode for 180 seconds over 10 trials, with no significant difference between the groups ($p = 0.196$) or trials ($p = 0.330$). Thus, we used the results obtained on day 3 as the baseline performance.

In the results for WT, a repeated-measures two-way ANOVA (EtOH dose × trial) of latency to fall showed a significant effect of EtOH dose (EtOH dose × trial, F(3, 299) = 21.056, $p < 0.001$), and a multiple comparison test suggested a significant decline in rotarod performance in the 1.6 g/kg EtOH injected group ($p < 0.05$) (Fig 3A). A one-way ANOVA of latency to fall in the same trials found a significant difference between the 1.6 g/kg EtOH injection and the saline in all three trials. On the other hand, in GAD65-KO, a two-way ANOVA (EtOH dose × trial) of latency to fall also detected a significant effect of EtOH injection (F(3, 279) = 11.727, $p < 0.001$), but a multiple comparison test suggested that a significant decline in rotarod performance occurred in the 1.2 and 1.6 g/kg EtOH-injected groups ($p < 0.05$) (Fig 3B). A one-way ANOVA of latency to fall in the same trials showed a significant decrease compared to the saline in trials 1 and 2 in the 1.2 g/kg EtOH-injected group and trials 1–3 in the 1.6 g/kg EtOH-injected group. Comparing WT and GAD65 KO, no significant difference in performance was observed at 0.8 and 1.6 g/kg EtOH. However, a significant difference was detected in the 1.2 g/kg EtOH-injected group by two-way ANOVA (F(1, 189) = 8.403, $p < 0.05$). These results suggest that EtOH, even at low concentrations, impairs motor coordination in GAD65-KO.

## Effects of EtOH on spontaneous firing of cerebellar PCs

Several previous studies reported that acute application of EtOH to PCs *in vitro* and *in vivo* has a tendency to facilitate spontaneous firing of PCs [37, 46, 47]. To examine the dose-dependent effects of EtOH on the firing rate of PCs in WT and GAD65-KO, we performed cell-attached recordings from PCs to minimize intracellular dialysis and applied EtOH at low and high doses. At first, we compared the basal firing rates of PCs which showed the continuous tonic firing mode between WT and GAD65-KO. We did not detect a significant difference in the basal firing rate of PCs we observed in this study (WT: 17.9 ± 1.8 Hz; n = 17; and GAD65-KO: 20.1 ± 2.2 Hz; n = 11, $p = 0.397$) (Fig 4A). Additionally, we examined the spike train regularity of PCs by measuring not only the CV but also CV2, which detects variability less sensitive to changes in the mean firing rate [42]. There was no significant difference between the genotypes in either CV (WT: 0.169 ± 0.016; n = 17; and GAD65-KO: 0.124 ± 0.016; n = 11, $p = 0.100$) or CV2 (WT: 0.170 ± 0.015; n = 17; and GAD65-KO: 0.136 ± 0.019; n = 11, $p = 0.121$) (Fig 4A). As shown in Fig 3B, only GAD65-KO showed low rotarod performance when they received a 1.2 g/kg EtOH injection. At this dose, the blood EtOH concentration 10 min after injection into WT was 0.238 ± 0.026% (Fig 1C), which was approximately 53 mM. On the other hand, the blood EtOH concentration after a 1.6 g/kg EtOH injection was 0.383 ± 0.067% (n = 3), which is approximately 85 mM. Thus, here, we bath-applied EtOH to cerebellar slices at concentrations of 50 mM and more than 100 mM [32]. EtOH (50 mM) facilitated PC firing significantly in not only WT (from 19.3 ± 2.3 to 22.1 ± 2.5 Hz; n = 8; $p < 0.05$) but also GAD65-KO (from 18.9 ± 1.4 to 25.1 ± 1.6 Hz; n = 5; $p < 0.05$) (Fig 4B–4D).

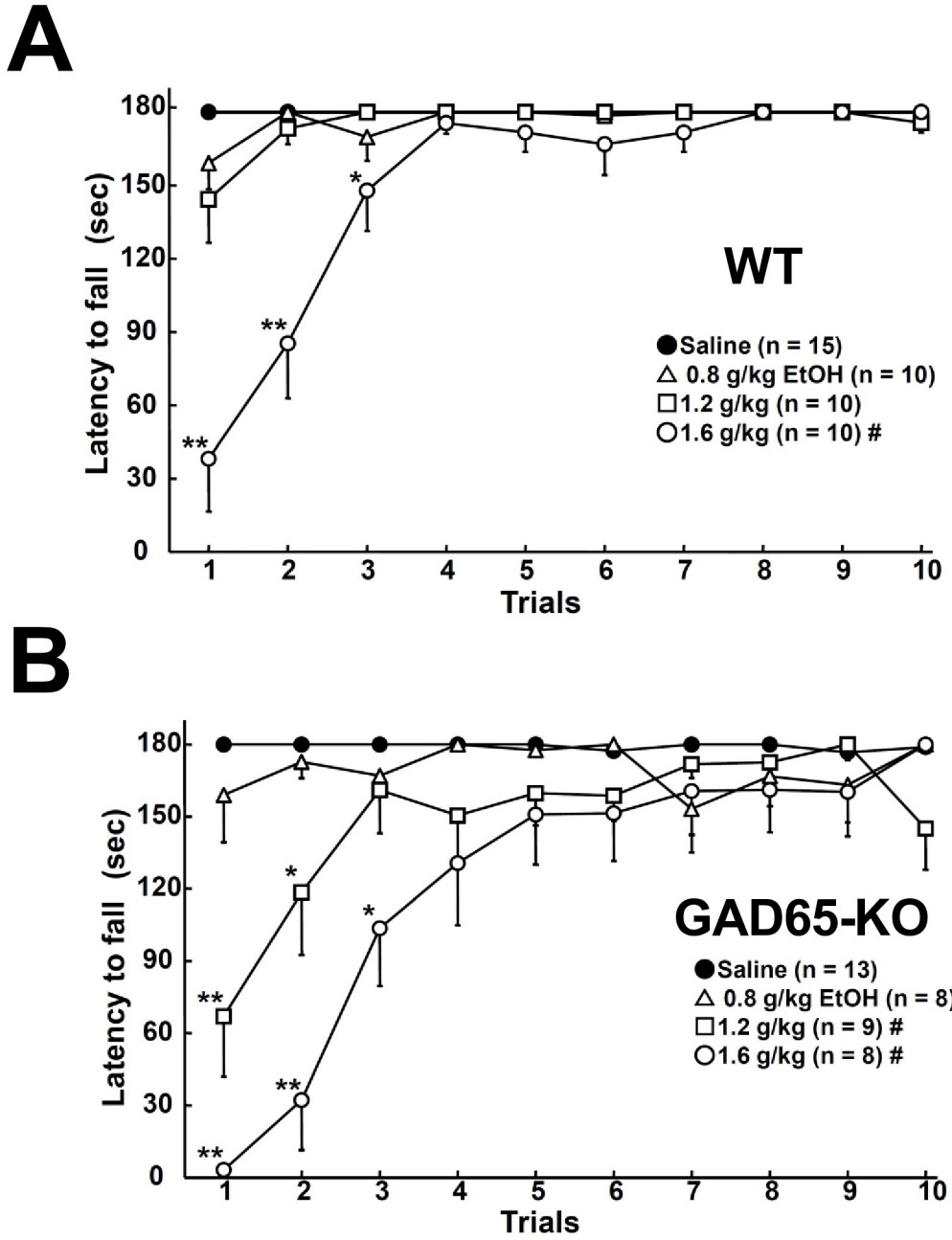

**Fig 3. Changes in rotarod performance after EtOH injection in WT and GAD65 KO.** In WT, a significant decline was observed in the 1.6 g/kg EtOH-injected group (A). On the other hand, GAD65-KO showed a significant decline with 1.2 and 1.6 g/kg EtOH injections (B). $^\#p < 0.05$: a significant difference compared to the saline-injected group detected by a two-way ANOVA. $^*p < 0.05$, $^{**}p < 0.01$: a significant difference compared to the saline-injected group within the same trial, detected by a one-way ANOVA.

The magnitude of the increase was higher in GAD65-KO (133 ± 6% of control) than in WT mice (116 ± 4% of control) ($p < 0.05$) (Fig 4D), suggesting that the EtOH-induced facilitation of PC firing at the low concentration was enhanced in GAD65-KO compared to WT. 50 mM EtOH did not alter the spike train regularity of PC firing in WT (CV: from 0.157 ± 0.021 to 0.141 ± 0.014; n = 8; $p = 0.441$; and CV2: from 0.173 ± 0.023 to 0.153 ± 0.014; n = 8; $p = 0.232$)

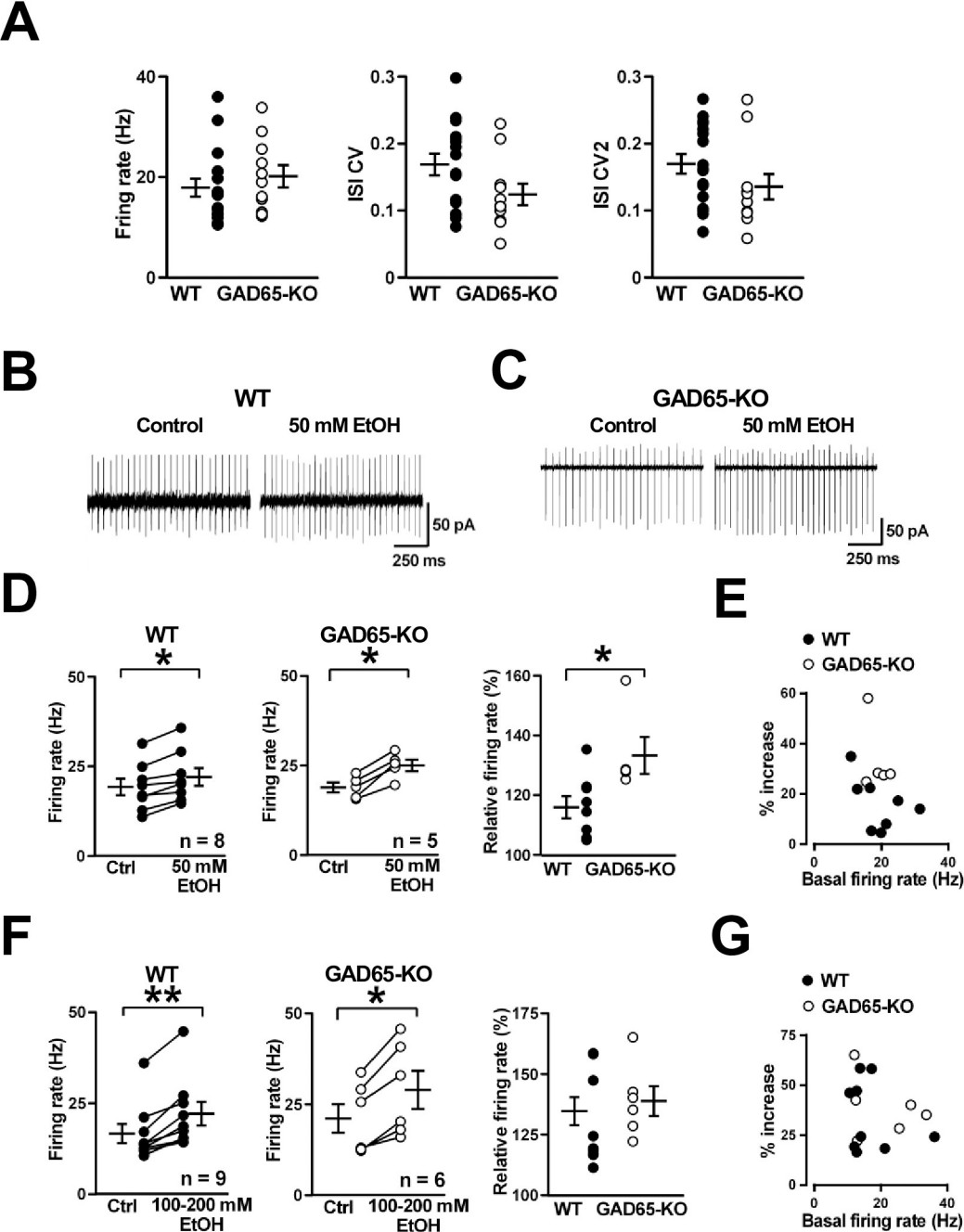

**Fig 4. Effects of EtOH on spontaneous firing of cerebellar PCs.** (A) Comparisons of spontaneous firing in PCs between WT (n = 17) and GAD65-KO (n = 11). There were no significant differences in the firing rate (left), coefficient of variation (CV) (middle), or CV2 (right) of the inter-spike interval. (B and C) Representative traces of spontaneous firing recorded by the cell-attached mode from PCs before treatment (control) and in the presence of EtOH (50 mM) in WT (B) and GAD65-KO (C). (D) 50 mM EtOH increased the firing rate of PCs significantly in WT (n = 8, left) and GAD65-KO (n = 5, middle). The magnitude of the increase was higher in GAD65-KO than in WT (right). (E) Relationship between the 50 mM EtOH-induced % increase and basal firing rate of PCs. (F) High-dose EtOH (100–200 mM) facilitated PC firing in WT (n = 9, left) and GAD65-KO (n = 6, middle). There was no difference in the magnitude of facilitation between WT and GAD65-KO (right). (G) Relationship between the high-dose EtOH-induced % increase and basal firing rate of PCs. $*p < 0.05$, $**p < 0.01$.

or GAD65-KO (CV: from 0.147 ± 0.030 to 0.171 ± 0.022; n = 5; $p$ = 0.106; and CV2: from 0.168 ± 0.036 to 0.198 ± 0.022; n = 5; $p$ = 0.281). The relationship between the 50 mM EtOH-induced % increase and basal firing rate of PCs were shown in Fig 4E. At high doses (100–200 mM), EtOH also increased the firing rate of PCs in WT (from 16.7 ± 2.6 to 22.2 ± 3.2 Hz; n = 9; $p < 0.01$) and GAD65-KO (from 21.2 ± 3.9 to 29.0 ± 5.2 Hz; n = 6; $p < 0.05$) (Fig 4F). No statistical difference was detected in the magnitude of increases between WT (135 ± 6% of control) and GAD65-KO (139 ± 6% of control) ($p$ = 0.556). Even at 100–200 mM, EtOH did not change the spike train regularity of PC firing in WT (CV: from 0.181 ± 0.023 to 0.175 ± 0.022; n = 9; $p$ = 0.953; and CV2: from 0.167 ± 0.020 to 0.188 ± 0.025; n = 9; $p$ = 0.286) or GAD65-KO (CV: from 0.106 ± 0.014 to 0.152 ± 0.036; n = 6; $p$ = 0.295, CV2: from 0.109 ± 0.012 to 0.152 ± 0.032; n = 6; $p$ = 0.344). The distributions of plots showing the relationship between the 100–200 mM EtOH-induced % increase and basal firing rate of PCs were similar in WT and GAD65-KO (Fig 4G). These results suggest that GAD65-KO are more sensitive than WT to the facilitating effect of EtOH on cerebellar PC firing.

## Discussion

We report here that motor coordination and locomotor activity in GAD65-KO are more sensitive to the effect of EtOH than WT, although there is no difference in blood EtOH concentration between the genotypes. Our *in vitro* experiments also show that the firing facilitation of cerebellar PCs induced by EtOH at 50 mM is enhanced in GAD65-KO compared to WT, but not at higher concentrations. These results suggest that GAD65-KO, which show the low GABA concentration in the cerebellum, impair suppressing the facilitation of PC firing and maintaining normal motor coordination during exposure to EtOH at the lower doses.

The home cage activity of GAD65-KO was lower than that of WT during the nighttime. In contrast, our open-field test showed that baseline locomotion in GAD65-KO was significantly higher than that in WT, which is consistent with previous studies [15, 18]. These different activities between the genotypes could be attributed to increased anxiety in GAD65-KO [15, 18]. The total distance moved by WT was not altered by EtOH treatment. WT injected with 1.6 g/kg EtOH showed an increase in locomotor activity during the first 5 min of the test period but were no different from the baseline in later time blocks. This tendency is similar to other studies on the locomotor stimulation effect of EtOH [23]. By contrast, GAD65-KO treated with 1.2 and 1.6 g/kg EtOH showed a significant increase in distance moved and higher locomotion than the baseline throughout the experimental period, suggesting that the KO mice are more sensitive to the locomotor stimulant effect of EtOH and that this effect is maintained for a longer period than WT. The alteration in locomotor activity is a well-known characteristic of alcohol-induced ataxia [48–50]. It is enhanced by EtOH dose-dependently but suppressed at its high doses. The threshold EtOH dose for locomotive stimulation varies by strain in mice [51, 52], whereas previous studies used more than 1.5 g/kg EtOH injection. EtOH doses we used here were lower for locomotion enhancement in WT. GAD65-KO showed higher open-field locomotor activity at baseline, and the locomotion stimulant effect of EtOH facilitated their basic locomotor activity. This result could be evidenced by previous studies showing that systemic administration of agonists for GABA receptors reduces the EtOH-induced locomotor stimulant response [53], whereas systemic administration of antagonists for GABA receptors enhances the locomotor stimulant response [45, 54, 55]. Thus, GABAergic inhibition participates in the suppression of EtOH-induced alteration of locomotor activity and motor coordination [6, 8, 28].

Although there was no difference in motor coordination or motor learning between GAD65-KO and WT in the rotarod test even in having acute intraperitoneally administered

injections of 2.0 g/kg EtOH [56], a 1.2 g/kg EtOH injection caused a significant decrease in rotarod performance in only GAD65-KO. Since impairment of motor coordination is one of the typical symptoms of acute EtOH exposure, we can consider that motor coordination in GAD65-KO is more sensitive to EtOH than that in WT. This is most likely due to the low GABA concentration in the brain of GAD65-KO [15, 17]. A previous study reported that EtOH-induced ataxia is caused by an enhancement of extrasynaptic GABA$_A$ receptor activity [8], but the mechanism underlying motor coordination is more complex [57]. It is possible that other neural functions which are also sensitive to low-dose EtOH [58] can contribute to the induction of ataxia in GAD65-KO exposed acutely to EtOH at the low concentrations. Acute administration of EtOH has various functions through its action on multiple sites in the cerebellum [31, 47]. Acute EtOH has been reported to facilitate neuronal firing via the activation of hyperpolarization-activated cyclic nucleotide-gated channels of GABAergic interneurons in the cerebellar molecular layers [32], and via the inhibition of KCNQ channels of dopamine neurons in the ventral tegmental area [59]. Since cerebellar PCs express these channels [60, 61], EtOH can excite PCs and facilitate their spontaneous firing. Additionally, according to literature showing that EtOH reduces voltage-gated Ca$^{2+}$ currents in cerebellar PCs [62] and that a decrease in P-type Ca$^{2+}$ channel currents facilitates PC firing [63], EtOH likely increases the firing rate of PCs by reducing voltage-gated Ca$^{2+}$ currents.

Our previous study indicated that blockade of GABAergic inhibition reduces the dose threshold of EtOH that facilitates the firing of cerebellar interneurons and PCs [32], suggesting that GABAergic transmission in the cerebellar cortex can keep the extent of the EtOH-mediated effects on these neuronal firing smaller especially at lower-doses of EtOH. When we recorded PC firing in the presence of synaptic blockers such as kynurenic acid and bicuculline, EtOH (50 mM) facilitated PC firing in WT (from 26.2 ± 2.8 to 33.5 ± 3.9 Hz; n = 6; $p < 0.05$; 127 ± 3% of control). Although we only had two cell data for GAD65-KO, extents of a 50 mM EtOH-induced increase in PC firing in GAD65-KO (from 19.0 to 23.8 Hz; 125% of control, and from 18.5 to 22.6 Hz; 122% of control) were similar to the averaged value of WT. It has been reported that the patterns of single spikes in PCs are modified by inhibitory GABAergic transmission rather than by excitatory synaptic transmission [64]. Taken together, we considered that the significant difference between WT and GAD65-KO, as shown in the right panel of Fig 4D, could have be caused by the suppression of EtOH-mediated facilitation of inhibitory GABAergic transmission in GAD65-KO. In GAD65-KO, the GABA content in the cerebellum is reduced to 75% of that in WT [15, 17]. Thus, the higher sensitivity of motor coordination to EtOH in GAD65-KO could depend on the low GABA concentration in the cerebellum, where GABAergic transmission onto PCs evoked by intermittent stimulation does not change significantly in GAD65-KO [17]. Furthermore, it is conceivable that the normal GABA concentration in the WT brains is able to protect neurons from overexcitation caused by EtOH exposure at low doses. Taken together, GAD65-KO have a higher sensitivity to the effects of acute exposure to EtOH than WT. This different sensitivity could be mainly associated with the GABA concentration in the brain. Thus, GAD65-KO seem to be useful as an animal model for elucidating the mechanism underlying acute alcohol-induced ataxia.

## Supporting information

**S1 Data. The numerical values underlying Fig 4.**
(XLSX)

## Acknowledgments

We thank Prof. Kunihiko Obata for his work on the initial draft of this paper, Dr. Yuchio Yanagawa for his critical reading of this manuscript and Editage (www.editage.com) for the English language review.

## Author Contributions

**Conceptualization:** Wataru Matsunaga.

**Data curation:** Wataru Matsunaga, Moritoshi Hirono.

**Formal analysis:** Wataru Matsunaga, Toru Shinoe, Moritoshi Hirono.

**Investigation:** Wataru Matsunaga, Moritoshi Hirono.

**Methodology:** Wataru Matsunaga, Toru Shinoe, Moritoshi Hirono.

**Project administration:** Wataru Matsunaga.

**Resources:** Wataru Matsunaga, Toru Shinoe.

**Software:** Toru Shinoe.

**Validation:** Wataru Matsunaga, Toru Shinoe, Moritoshi Hirono.

**Visualization:** Wataru Matsunaga, Moritoshi Hirono.

**Writing – original draft:** Wataru Matsunaga.

**Writing – review & editing:** Wataru Matsunaga, Toru Shinoe, Moritoshi Hirono.

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
