## [Decision Letter · Decision Letter 0]

20 Feb 2023

PONE-D-23-01915GAD65 deficient mice are susceptible to ethanol-induced impairment of motor coordination and facilitation of cerebellar neuronal firingPLOS ONE

Dear Dr. Hirono,

Thank you for submitting your manuscript to PLOS ONE. After careful consideration, we feel that it has merit but does not fully meet PLOS ONE’s publication criteria as it currently stands. Therefore, we invite you to submit a revised version of the manuscript that addresses the points raised during the review process.

We look forward to receiving your revised manuscript.

Kind regards,

Yuqing Li, Ph.D.

Academic Editor

PLOS ONE

Reviewers' comments:

Reviewer's Responses to Questions

**Comments to the Author**

1. Is the manuscript technically sound, and do the data support the conclusions?

Reviewer #1: Partly

Reviewer #2: Yes

Reviewer #3: Yes

2. Has the statistical analysis been performed appropriately and rigorously? 

Reviewer #1: I Don't Know

Reviewer #2: Yes

Reviewer #3: Yes

3. Have the authors made all data underlying the findings in their manuscript fully available?

Reviewer #1: Yes

Reviewer #2: Yes

Reviewer #3: Yes

4. Is the manuscript presented in an intelligible fashion and written in standard English?

Reviewer #1: Yes

Reviewer #2: Yes

Reviewer #3: Yes

5. Review Comments to the Author

Reviewer #1: The manuscript investigated motor coordination and cerebellar Purkinje cell spontaneous activity in GAD65 KO mice with EtOH. It is very interesting to see a higher sensitivity of GAD65 KO to the effect of EtOH. Overall, the idea is novel and the study is designed well. The following are some concerns about the results:

1. Line 83, would you mind providing PCR primers used for genotyping? Are the WT and GAD65 KO used in the experiment littermates?

2. For PC recording, were inhibitory/excitatory synaptic transmission blocked? How to rule out the influence from other cells and the circuit if no blocking? How much the cell/internal resistance was? Internal resistance has impact on flow of current therefore is important to see if constant.

3. For PC recording, did you distinguish tonic and non-tonic cells? From representative figures, it seems to be tonic cell. Do you have representative traces for non-tonic cell? How was the regularity calculated? Please provide more details about CV and CV2 measurements. Additionally, it will be very interesting if more electrophysiology work, like current step, can be done.

4. Line 202, would you mind disclosing which previous study this is? Please cite the paper accordingly.

5. Figure 1C, would you mind providing more rationale about the reason to measure EtOH 10min, 30min and 60min after injection? For open field test, the mice were put in 3min post dose and the study lasted for 30min. Is it more meaningful to measure blood EtOH concentration predose, 3 min, 10min and 30 min? Additionally, is there any reason for only plotting data with 1.2 g/kg EtOH, but not 1.6 g/kg?

6. Please provide more information about how Figure 1A was made and how statistics were done for activity comparison. Does each dot in the figure represent averaged activity from all animals and all days?

Reviewer #2: In this paper, the authors describe the sensitivity of motor coordination and the spontaneous firing of cerebellar Purkinje cells to EtOH are attributed to basal GABA concentration in the brain using conventional GAD65-KO mice. Although overall data are solid and support the hypothesis, a number of issues should be addressed before publication, as described below. I would recommend the paper for publication after thorough revision.

Major points:

1. Line 276: The authors should add the detailed analysis procedure of CV2 to the Materials and Methods section, not just by citing paper #45.

2. Figure 1A,B: GAD65-KO mice have less activities in the home cage than wild-type mice (A). However, GAD65-KO mice have more activities in the open field test than wild-type mice (B). The authors need to reconcile the contradictory data between them.

3. Figure 4C,D: The data variability of basal firing rate of GAD65 KO mice is different between C and D, even though the experimental conditions should be the same. Similarly, the average values are different in the basal condition of wild-type mice between C (19.7 +/- 3.2) and D (16.7 +/- 2.6). The effects of EtOH on changes in firing rate could be affected to the basal firing rate of the cell that are selected by chance or to the cell condition of the slice. It is very important to confirm that the same reported findings are obtained even if conditions are standardized.

Reviewer #3: In this paper, the authors examined whether motor coordination and locomotor activity in GAD65-KO are more sensitive to the effect of EtOH than in WT. They performed behavioral tests and electrophysiological experiments. Their experiments are well designed and documented in the text. However, interpretation of the data is not sufficient.

(1) Spontaneous firing of Purkinje cells has been shown to increase with ethanol exposure in both WT and GAD65KO. What is the possible mechanism for this?　

(2) Ethanol sensitivity has been shown to be increased in GAD65KO compared to WT. What is the possible mechanism for this?

6. PLOS authors have the option to publish the peer review history of their article (what does this mean?). If published, this will include your full peer review and any attached files.

Reviewer #1: No

Reviewer #2: No

Reviewer #3: No

---

## [Author Response · Author response to Decision Letter 0]

5 Apr 2023

>>Reviewer #1: The manuscript investigated motor coordination and cerebellar Purkinje cell spontaneous activity in GAD65 KO mice with EtOH. It is very interesting to see a higher sensitivity of GAD65 KO to the effect of EtOH. Overall, the idea is novel and the study is designed well. The following are some concerns about the results:

We thank the reviewer for these valuable comments and suggestions.

>>1. Line 83, would you mind providing PCR primers used for genotyping? Are the WT and GAD65 KO used in the experiment littermates?

As shown in the following link, The RIKEN standard protocol was used for genotyping. We added this information in the Materials and Methods section (Page 4, Line 84), “https://mus.list.brc.riken.jp/ja/wp-content/pdf/00989_PCR.pdf”. As described by the reviewer, we used littermates for the experiments. This information was also added in the Materials and Methods section (Page 4, Lines 80–81).

>>2. For PC recording, were inhibitory/excitatory synaptic transmission blocked? How to rule out the influence from other cells and the circuit if no blocking? How much the cell/internal resistance was? Internal resistance has impact on flow of current therefore is important to see if constant.

Given this recording condition, synaptic transmission remained uninhibited. The present study provides evidence that the higher sensitivity of EtOH for motor coordination in GAD65-KO (Fig. 3B) could have depended on the attenuation of GABAergic inhibition of EtOH-induced facilitation of PC firing (revised Fig. 4B-E) resulting from low GABA concentration in the cerebellum. We recorded PC firing in the presence of synaptic blockers i.e., kynurenic acid (1 mM) and bicuculline (10 μM). In this condition, EtOH (50 mM) facilitated PC firing in WT (from 26.2 ± 2.8 to 33.5 ± 3.9 Hz; n = 6; p < 0.05; 127 ± 3% of control). Unfortunately, we only had two cell data for GAD65-KO i.e., EtOH facilitated PC firing (from 19.0 to 23.8 Hz; 125% of control, and from 18.5 to 22.6 Hz; 122% of control) because of limited GAD65-KO supply. According to these underlined values, WT and GAD65-KO showed similar extents of a 50 mM EtOH-induced increase in the firing rate when synaptic transmission was blocked. It has been reported that the patterns of single spikes in PCs are modified by inhibitory GABAergic synaptic transmission rather than by excitatory synaptic transmission (Häusser and Clark, 1997 Neuron). Taken together, we considered that the significant difference between WT and GAD65-KO, as shown in the right panel of revised Fig. 4D, could have be caused by the suppression of EtOH-mediated facilitation of inhibitory GABAergic transmission in GAD65-KO.

As the reviewer suggested, we agree that it is important to evaluate whether EtOH can affect the membrane resistance of PCs and whether it is significantly different between WT and GAD65-KO. In this study, however, we applied only cell-attached recordings to PCs which minimize intracellular dialysis and prevent the rundown of fragile channel currents, because we sought to observe changes in PC firing resulting from the total and multiple effects of EtOH on PCs.

>>3. For PC recording, did you distinguish tonic and non-tonic cells? From representative figures, it seems to be tonic cell. Do you have representative traces for non-tonic cell? How was the regularity calculated? Please provide more details about CV and CV2 measurements. Additionally, it will be very interesting if more electrophysiology work, like current step, can be done.

Thank you for the reviewer’s valuable comments.

The firing patterns of PCs are generally divided into three types: tonic, bursting, and silent (Womack and Khodakhah, 2002 J Neurosci). We bath-applied EtOH to PCs that exhibited continuous tonic firing patterns to estimate the effect of EtOH on PC excitability upon increasing firing rate. Previous studies have reported that PCs in slices and in vivo fire tonically, with occasional periods of bursting and silence. In our cerebellar slice preparations, most of the PCs fired tonically, while some were silent. This may have been due to the damage to their axon initial segments during slice cutting. Only a few of the PCs fired in the bursting mode. Unfortunately, we did not obtain representative traces of non-tonic cells.

As the reviewer suggested we further elaborated on CV2 in the Materials and Methods section (Page 8, Lines 188–193). 

We sincerely apologize that we miswrote some values of the firing rates, CV, CV2 and p in the original Result section. We have corrected them in red (Page 12–13, Lines 291–308). There is no change in their conclusions.

As the reviewer suggested, it is important to compare and elucidate the electrophysiological properties of PCs between GAD65-KO and WT in addition to synaptic transmission onto PC (Obata et al., 1999 Neuroscience). Further studies are needed to find out which molecules are affected by GAD65-deletion and involved in changing EtOH sensitivity. Additionally, we have a limitation for GAD65-KO, thus, hindering the execution of these experiments.

>>4. Line 202, would you mind disclosing which previous study this is? Please cite the paper accordingly.

We cited that of Kitanaka and Ueno (1975, J Jap Psychosom Soc) (Page 9, Line 214).

>>5. Figure 1C, would you mind providing more rationale about the reason to measure EtOH 10min, 30min and 60min after injection? For open field test, the mice were put in 3min post dose and the study lasted for 30min. Is it more meaningful to measure blood EtOH concentration predose, 3 min, 10min and 30 min? Additionally, is there any reason for only plotting data with 1.2 g/kg EtOH, but not 1.6 g/kg?

This experiment was conducted to examine whether 'time to sobriety' is different between WT and GAD65-KO. This experiment is not related to the open field test. We added an expression (Page 10, Lines 225–226).

>>6. Please provide more information about how Figure 1A was made and how statistics were done for activity comparison. Does each dot in the figure represent averaged activity from all animals and all days?

The data in Fig. 1A were calculated as follows: (1) averaging the activity of each mouse every 60 mins over the entire measurement period, (2) grouping the data from each mouse into WT and GAD65-KO, and (3) averaging them. Statistical analyses between WT and KO were performed by two-way analysis of variance for day and night separately.

Thus, as the reviewer pointed out, we have added more descriptions in the Results section (Page 9, Lines 197–207) and in the Figure Legends of Fig. 1 (Page 22, Lines 578–580).

>>Reviewer #2: In this paper, the authors describe the sensitivity of motor coordination and the spontaneous firing of cerebellar Purkinje cells to EtOH are attributed to basal GABA concentration in the brain using conventional GAD65-KO mice. Although overall data are solid and support the hypothesis, a number of issues should be addressed before publication, as described below. I would recommend the paper for publication after thorough revision.

We thank the reviewer for these valuable comments and suggestions.

>>Major points:

>>1. Line 276: The authors should add the detailed analysis procedure of CV2 to the Materials and Methods section, not just by citing paper #45.

As the reviewer suggested, we have added a detailed description to the Materials and Methods section (Page 8, Lines 188–193) as we have already answered Comment 3 of Reviewer#1.

>>2. Figure 1A,B: GAD65-KO mice have less activities in the home cage than wild-type mice (A). However, GAD65-KO mice have more activities in the open field test than wild-type mice (B). The authors need to reconcile the contradictory data between them.

Locomotion activity in an unfamiliar open field cannot be equated with activity in a familiar home cage. It includes elements of exploratory and anxiety behaviors in a new environment. Thus, we added sentences in the Discussion section (Page 14, Lines 323–327).

>>3. Figure 4C,D: The data variability of basal firing rate of GAD65 KO mice is different between C and D, even though the experimental conditions should be the same. Similarly, the average values are different in the basal condition of wild-type mice between C (19.7 +/- 3.2) and D (16.7 +/- 2.6). The effects of EtOH on changes in firing rate could be affected to the basal firing rate of the cell that are selected by chance or to the cell condition of the slice. It is very important to confirm that the same reported findings are obtained even if conditions are standardized.

Regarding GAD65-KO and its insignificant difference in the basal firing rate between the original Fig. 4C (18.9 ± 1.4 Hz; n = 5) and D (21.2 ± 3.9 Hz; n = 6) (p = 0.536), there is a significant difference in its variance of basal firing rate between the original Fig. 4C (9.29; n = 5) and D (92.5; n = 6) (p = 0.045 by F-test). Although the exact reason remains unknown, this result may be attributable to the small amount of data. At present, the amount of data cannot be increased because the GAD65-KO are not maintained. On the other hand, regarding WT and its insignificant difference in the basal firing rate between the original Fig. 4C (19.3 ± 2.3 Hz; n = 8) and D (16.7 ± 2.6 Hz; n = 9) (p = 0.268 by F-test), there was no significant difference in its variance of basal firing rate between the original Fig. 4C (44.1; n = 8) and D (63.1; n = 9) (p = 0.650 by F-test).　In this study, one of the aims was to compare the properties of PC firing between WT and GAD65-KO. Thus, we have added distribution graphs for the firing rate, CV and CV2, as shown in revised Fig. 4A, and their descriptions in the Results section (Page 12, Lines 276–284).

To show the relationship between the effects of EtOH on changes in firing rate and the basal firing rate of the cell, we have added graphs for the basal firing rate vs. % increase of 50 mM EtOH and of 100–200 mM EtOH (revised Figs. 4E and 4G, respectively), and their descriptions in the Results section (Page 13, Lines 299–300 and 308–310).

We added explanations in the Figure Legends for revised Fig. 4 (Page 23, Lines 606–617).

>>Reviewer #3: In this paper, the authors examined whether motor coordination and locomotor activity in GAD65-KO are more sensitive to the effect of EtOH than in WT. They performed behavioral tests and electrophysiological experiments. Their experiments are well designed and documented in the text. However, interpretation of the data is not sufficient.

We thank the reviewer for these valuable comments and suggestions.

>>(1) Spontaneous firing of Purkinje cells has been shown to increase with ethanol exposure in both WT and GAD65KO. What is the possible mechanism for this?

We have added the possible mechanisms in the Discussion section (Page 15, Lines 357–365).

>>(2) Ethanol sensitivity has been shown to be increased in GAD65KO compared to WT. What is the possible mechanism for this?

We have described this possible mechanism in the Discussion section of the original manuscript. In the revised manuscript, we have modified several parts of the sentences in red (Page 15-16, Lines 365–373).

---

## [Decision Letter · Decision Letter 1]

2 May 2023

PONE-D-23-01915R1GAD65 deficient mice are susceptible to ethanol-induced impairment of motor coordination and facilitation of cerebellar neuronal firingPLOS ONE

Dear Dr. Hirono,

Thank you for submitting your manuscript to PLOS ONE. After careful consideration, we feel that it has merit but does not fully meet PLOS ONE’s publication criteria as it currently stands. Therefore, we invite you to submit a revised version of the manuscript that addresses the points raised during the review process.

We look forward to receiving your revised manuscript.

Kind regards,

Yuqing Li, Ph.D.

Academic Editor

PLOS ONE

Journal Requirements:

Additional Editor Comments:

Thank you for the extensive revision. Only a few minor issues need to be addressed from reviewer #1's comments.

Reviewers' comments:

Reviewer's Responses to Questions

**Comments to the Author**

1. If the authors have adequately addressed your comments raised in a previous round of review and you feel that this manuscript is now acceptable for publication, you may indicate that here to bypass the “Comments to the Author” section, enter your conflict of interest statement in the “Confidential to Editor” section, and submit your "Accept" recommendation.

Reviewer #1: (No Response)

Reviewer #3: All comments have been addressed

2. Is the manuscript technically sound, and do the data support the conclusions?

Reviewer #1: Yes

Reviewer #3: Yes

3. Has the statistical analysis been performed appropriately and rigorously? 

Reviewer #1: Yes

Reviewer #3: Yes

4. Have the authors made all data underlying the findings in their manuscript fully available?

Reviewer #1: Yes

Reviewer #3: Yes

5. Is the manuscript presented in an intelligible fashion and written in standard English?

Reviewer #1: Yes

Reviewer #3: Yes

6. Review Comments to the Author

Reviewer #1: I appreciate the authors' response to my questions. It is interesting to know that with synaptic blockers, the fold change of firing rate in WT and GAD65-KO are similar. Would you mind adding this in the manuscript and discuss the assumptions as in the response? Additionally, I hope the message that the experiments were done in tonic cells only can be emphasized since tonic/no-tonic cells may behave very differently. I appreciate that the authors mentioned this in line 277, but would you mind talking about this more in either method or discussion with Womack and Khodakhah, 2002 J Neurosci paper? Thank you very much again.

Reviewer #3: (No Response)

7. PLOS authors have the option to publish the peer review history of their article (what does this mean?). If published, this will include your full peer review and any attached files.

Reviewer #1: No

Reviewer #3: No

---

## [Author Response · Author response to Decision Letter 1]

4 May 2023

>>Journal Requirements:

>>Please review your reference list to ensure that it is complete and correct. If you have cited papers that have been retracted, please include the rationale for doing so in the manuscript text, or remove these references and replace them with relevant current references. Any changes to the reference list should be mentioned in the rebuttal letter that accompanies your revised manuscript. If you need to cite a retracted article, indicate the article’s retracted status in the References list and also include a citation and full reference for the retraction notice.

As the Journal Requirements suggested, we have checked our reference list. In this revised manuscript, we have added two articles (new [41] and new [64]) to respond to the Reviewer#1’s comments.

>>6. Review Comments to the Author

>>Reviewer #1: I appreciate the authors' response to my questions. It is interesting to know that with synaptic blockers, the fold change of firing rate in WT and GAD65-KO are similar. Would you mind adding this in the manuscript and discuss the assumptions as in the response? Additionally, I hope the message that the experiments were done in tonic cells only can be emphasized since tonic/no-tonic cells may behave very differently. I appreciate that the authors mentioned this in line 277, but would you mind talking about this more in either method or discussion with Womack and Khodakhah, 2002 J Neurosci paper? Thank you very much again.

We thank the reviewer for these valuable comments and suggestions.

As the reviewer suggested, we added descriptions of the data with synaptic blockers and the assumption in the Discussion section (Page 16, Lines 377–386). We also added descriptions of the firing patterns of PCs while citing the paper (Pages 7–8, Lines 169–176).

---

## [Decision Letter · Decision Letter 2]

7 May 2023

GAD65 deficient mice are susceptible to ethanol-induced impairment of motor coordination and facilitation of cerebellar neuronal firing

PONE-D-23-01915R2

Dear Dr. Hirono,

We’re pleased to inform you that your manuscript has been judged scientifically suitable for publication and will be formally accepted for publication once it meets all outstanding technical requirements.

Kind regards,

Yuqing Li, Ph.D.

Academic Editor

PLOS ONE

Additional Editor Comments (optional):

Reviewers' comments:

Reviewer's Responses to Questions

**Comments to the Author**

1. If the authors have adequately addressed your comments raised in a previous round of review and you feel that this manuscript is now acceptable for publication, you may indicate that here to bypass the “Comments to the Author” section, enter your conflict of interest statement in the “Confidential to Editor” section, and submit your "Accept" recommendation.

Reviewer #1: All comments have been addressed

2. Is the manuscript technically sound, and do the data support the conclusions?

Reviewer #1: (No Response)

3. Has the statistical analysis been performed appropriately and rigorously? 

Reviewer #1: (No Response)

4. Have the authors made all data underlying the findings in their manuscript fully available?

Reviewer #1: (No Response)

5. Is the manuscript presented in an intelligible fashion and written in standard English?

Reviewer #1: (No Response)

6. Review Comments to the Author

Reviewer #1: (No Response)

7. PLOS authors have the option to publish the peer review history of their article (what does this mean?). If published, this will include your full peer review and any attached files.

Reviewer #1: No

---

## [Editor Report · Acceptance letter]

12 May 2023

PONE-D-23-01915R2 

GAD65 deficient mice are susceptible to ethanol-induced impairment of motor coordination and facilitation of cerebellar neuronal firing 

Dear Dr. Hirono:

I'm pleased to inform you that your manuscript has been deemed suitable for publication in PLOS ONE. Congratulations! Your manuscript is now with our production department. 

Kind regards, 

on behalf of

Dr. Yuqing Li 

Academic Editor

PLOS ONE